# Path Planning for Delivery Robots Based on an Improved Ant Colony Optimization Algorithm Combined with Dynamic Window Approach

**DOI:** 10.3390/s26010072

**Published:** 2025-12-22

**Authors:** Limin Huang, Tao Hu, Jiabao Wei, Yifeng Guo, Xubin Tong, Jiaxin Ding, Hao Yang, Bin Zhong

**Affiliations:** 1School of Mechanical Engineering, Chengdu University, Chengdu 610106, China; hlmsn155@163.com (L.H.); hutao202500@163.com (T.H.); txb15158382526@163.com (X.T.);; 2Multipurpose Utilization of Mineral Resources, Chinese Academy of Geological Sciences, Chengdu 610041, China

**Keywords:** food delivery robot, path planning, ant colony optimization algorithm, dynamic window approach, algorithm combination

## Abstract

In meal delivery robot path planning, enabling the robot to find an optimal path that avoids obstacles within its workspace is a crucial step. Usually, the traditional ant colony optimization (ACO) suffers from slow convergence and blind search behavior in path planning, lacking dynamic obstacle avoidance functionality. Meanwhile, the dynamic window approach (DWA) tends to become entrapped in local optima during local path planning. It is therefore proposed that a hybrid path planning algorithm be developed, based on an improved IACO and DWA algorithm. To address issues such as aimless search, slow convergence speed, and low path smoothness in ACO, the concept of gravity from gravity search algorithms is introduced to direct the search. The acceleration of convergence is achieved through the implementation of path sorting and the administration of additional pheromone to superior paths in pheromone updates. The transition paths are optimized to address the issue of excessive path transitions in ACO, resulting in smoother paths. The key nodes of the obtained globally optimal path are used as local target points, serving as multiple target points for DWA operation to enable dynamic obstacle avoidance. Simulation results indicate that compared to the ACO, the IACO reduces path length by up to 30.03% and decreases path turns by up to 71.43% in four different static maps. In other static comparison experiments, the IACO demonstrated superior performance compared to the other tested algorithms. In dynamic experiments, the proposed fusion algorithm can plan smooth paths that successfully avoid both static and dynamic obstacles.

## 1. Introduction

Due to the rapid advancement of intelligent technologies, robotics has found applications across multiple domains [1,2]. Currently, the most prevalent are service robots, capable of performing tasks beneficial to humans. As a subset of service robots, delivery robots are increasingly deployed in venues such as hotels, restaurants, and shopping centers. Given the complexity of restaurant environments, path planning technology has emerged as pivotal element in enabling delivery robots to navigate to their destinations [3]. Path planning algorithms for delivery robots are primarily categorized into two types: global path planning and local path planning [4]. During operation, delivery robots require not only effective global path planning to determine optimal routes but also local path planning with obstacle avoidance capabilities to navigate dynamically changing obstacles.

In robot path planning, common algorithms employed for global path planning include the A* algorithm [5], rapidly exploring random tree (RRT) [6], the genetic algorithm (GA) [7], sparrow search algorithm (SSA) [8], and ant colony optimization (ACO) [9]. These algorithms each possess distinct advantages and are widely applied.

Among them, ACO, as a distributed intelligent bionic algorithm that simulates the mechanism of ants to find the shortest path between nest and food through pheromone communication, is widely cited in various fields. The ACO is robust because of its positive feedback mechanism and also has powerful global search capabilities. However, the ACO suffers from problems such as low search efficiency, slow convergence, and a tendency to fall into local optimization. In order to solve these problems, numerous researchers have extensively studied the ACO. Wu et al. and Li et al. both transformed the path planning problem into a multi-objective optimization problem through multiple comprehensive evaluation metrics. They, respectively, employed the farthest-point optimization strategy and multi-optimization strategies (such as non-uniform initial pheromone distribution and ε-greedy algorithms) to achieve more comprehensive path optimization [10,11]. Wu et al. advanced a novel variant of the ACO, which embeds information such as directional information and path turns into an enhanced heuristic function, improving the quality of paths and search efficiency [12]. Zhang et al. developed an adaptive ant colony algorithm based on population information entropy, while Gao et al. incorporated backtracking and path merging strategies. Both approaches employed similar pheromone diffusion mechanisms for pheromone refinement, achieving a balance between path exploration and convergence speed or enhancing global search capabilities [13,14]. Cui et al. used deterministic state transfer probability rules to accelerate the convergence speed of the ACO and adaptive heuristic functions to optimize the number of turns and path length [15]. Liang et al. applied an ACO to global path planning for deep-sea mining vehicles, while introducing a pheromone updating strategy with a reward-penalty mechanism and an angle-inspired state transfer probability to make the generated paths smoother [16]. Liu et al. enhanced the guidance of pre-selected nodes during the initial search process through a pheromone concentration adaptive setting mechanism and also introduced a pseudo-random transfer strategy and dynamically adjusted pheromone volatilization rate to increase the population diversity and strengthen global search capability [17]. Fu et al. designed a grid environment model based on bidirectional artificial potential fields and introduced potential fields into the environment to provide direction for the ants. They also used the potential energy difference between nodes to optimize the pseudo-random state transfer rule. This improvement enhanced the efficiency of path planning and avoided blind search [18].

In addition to the above algorithms, there are a range of other effective path planning methods. For example, the A*, as a classical global path planning algorithm for static environments, has the advantages of being flexible, efficient, and generating relatively optimal paths, and is often used for path planning of mobile robots. Huang et al. proposed a 5-domain search method with the objective of improving efficiency and ensuring path smoothness [19]. Xu et al. used exploration from both the starting point and the goal point to reduce the number of nodes to be explored, and also improved the A* algorithm by designing a new adaptive cost function and a Slide-Rail corner adjustment method to improve path security and smoothness [20]. The RRT algorithm is a probabilistically complete algorithm for constructing a search tree by random sampling. Ganesan et al. solved the problems of slow convergence with uniform sampling and restricted exploration with non-uniform sampling by utilizing a path planning method with a mixed sampling RRT* that uses both non-uniform and uniform samplers [21]. Li et al. proposed a knowledge-based GA with a novel path representation combining grids and coordinates and a new evaluation method to make it better adapted to very complex environments [22]. The improved same-neighbor crossover operator and fitness function proposed by Lamini et al. for solving GA path planning problems in static environments resulted in a better average number of iterations and average number of transitions [23]. To overcome the susceptibility of SSA to fall into local optima and slow convergence speed, Yan et al. proposed an improved SSA with better search accuracy and faster convergence speed in mobile robot path planning [24].

The local path planning of the food delivery robot needs real-time adjustments according to the data collected by the sensors to ensure that the obstacle avoidance function can be achieved in complex environments. The commonly used local path planning algorithms are artificial potential field (APF) and dynamic window approach (DWA). Researchers enhance the performance of these algorithms by improving them. Liu et al. designed an improved method based on an environment-aware model for RRT guidance, which solves the problems of local minima, target inaccessibility, as well as local trajectory oscillations generated by the APF [25]. In order to address the problem that traditional APF methods are prone to generating zigzag paths in self-driving vehicles, Li et al. designed a real-time path planning method for self-driving cars using the dynamically enhanced fireworks algorithm APF [26]. Shin et al. considered localization accuracy in path planning and proposed a hybrid use of potential and localization risk fields to generate hybrid directed flows to guide driverless vehicles safely and efficiently [27]. The DWA algorithm is also used as a classical and practical local path planning algorithm, which is applied in several fields owing to its strong applicability. Yao et al. advanced a fuzzy logic improved DWA to address the problem of poor robustness of the DWA as well as the non-smoothness of the path, which achieves better path length, smoothness and robustness [28]. Zhang et al. improved the security and stability of local planning by introducing path smoothing coefficients and an improved DWA algorithm with a local goal selection strategy in path planning for USVs [29]. When applying DWA to mobile robot formations, Cao et al. implemented several measures to enhance navigational safety. These included analyzing formation patterns and surrounding obstacle environments, evaluating velocity change coefficients, and designing safety obstacle avoidance distance evaluation coefficients [30]. Wang et al. enabled agricultural robots to navigate more safely and autonomously in complex agricultural environments by integrating a dual-delay depth deterministic strategy gradient (TD3) into DWA and introducing an obstacle motion estimation module [31]. Wang et al. designed a distributed optimization method and an event-triggered behavior switching strategy to enable UGV formations to harmonize formation maintenance, global navigation, and local obstacle avoidance [32]. Abubakr et al. proposed an objective function weight that can optimize the DWA, considering the dynamic characteristics of the obstacles and the use of a fuzzy logic control system that can move quickly toward the target [33]. Niu et al. enhanced the DWA algorithm by using the data acquired from high-precision sensors on the vehicle and utilizing an ACO to update the speed objective function in real time to address the DWA algorithm’s problem of irrational path planning and incapacity to balance speed and driving safety during the task of crossing substantial obstacles [34]. Zhong et al. proposed a hybrid path planning algorithm based on an improved DWA, which enhances the obstacle avoidance effect and real-time path optimization of the DWA algorithm by using the A* algorithm to plan paths and avoid static obstacles in the operating environment, as well as by designing a target selection strategy and an improved method incorporating the speed barrier method [35].

Figure 1 illustrates the general workflow of path planning integrated with control. For a mobile robot to accomplish a navigation process, it generally requires a closed-loop sequence involving perception, positioning, planning, and control. The robot moves from the starting point to the target point while avoiding obstacles and ensuring the optimal path. The perception module acquires the robot’s surrounding environment and its current pose; Path planners are employed to identify optimal routes. Hybrid path planning integrates the global optimization capabilities of global planners with the dynamic responsiveness of local planners, thereby achieving the dual objectives of global optimality and local safety. The execution module will perform real-time motion based on the path planner’s output (linear and angular velocities), while simultaneously feeding back the motion status and pose to the perception module and local planner, thereby forming a closed-loop control system.

The global and local path planning algorithms described in the aforementioned literature each exhibit certain limitations. Scene adaptability is limited, with most improved algorithms designed for specific environments (such as static grids), exhibiting insufficient generalization capabilities in complex dynamic scenarios featuring multiple moving obstacles and abrupt terrain changes. In environments with dense dynamic obstacles, there is insufficient integrated capability regarding real-time performance, safety, and path smoothness. This paper proposes an approach that combines an improved ACO with a DWA to address the constraints in both global and local path planning for delivery robots. The main contributions are as follows:(1)Improvements to the ACO employ the following strategies: A distribution strategy for initial pheromone concentration is determined by integrating node location data with obstacle information. New heuristics are constructed by incorporating a gravitational mechanism and directional angle heuristics to refine state transition probability rules. Pheromone update rules are implemented by sorting paths and augmenting superior paths with additional pheromone. Simultaneously, redundant turning points in paths are optimized to enhance path smoothness.(2)Key nodes in global path planning are employed as target points for local paths. Dynamic obstacle distance evaluation subfunctions, linear velocity change evaluation subfunctions, and path proximity evaluation subfunctions are introduced. These are integrated with an IACO.(3)Experimental verification was conducted on both the IACO and the fusion algorithm. Simulation experiments demonstrated that the IACO exhibits superior advantages in path length, smoothness, and convergence time. In obstacle avoidance experiments, the fusion algorithm effectively achieves obstacle avoidance planning.

The content of this paper is as follows: Section 2 introduces the fundamental theory of traditional ACO and IACO algorithms. Section 3 describes enhancements to the DWA and the overall rules for the hybrid algorithm. Section 4 details simulation experiments and result analysis. Section 5 presents the conclusions.

## 2. An Improved ACO (IACO) for Global Path Planning

### 2.1. Establish a Grid-Based Environmental Model

Before applying path planning algorithms to robotic path planning, the surrounding environment must be abstracted into a mathematical model. For two-dimensional environments, using the grid method for modeling is a more widely adopted and effective approach. In grid maps, the surrounding environment is treated as a two-dimensional plane, which is divided into an array of squares of equal size containing binary information. Typically, black squares represent obstacles or impassable areas, with a value of 1; white squares represent passable areas, with a value of 0. Each grid has a corresponding serial number, which is sorted sequentially from left to right and top to bottom, as shown in Figure 2.

For grid maps with coordinates, each grid pixel has unique coordinates. Assuming the grid has $M_{x}$ rows and $M_{y}$ columns, the position of the *i-th*. grid pixel is obtained using the following Equation (1):(1)$$\left\{\begin{array}{c} x_{i}=a\left[\mathrm{mod}\left(i,M_{y}\right)-a/2\right]\\ y_{i}=a\left[x+a/2-\mathrm{ceil}\left(i/M_{x}\right)\right] \end{array}\right.$$
where a denotes the side length of the small square; mod denotes the modulo operation; ceil denotes the ceiling function.

### 2.2. Traditional Ant Colony Optimization

ACO was proposed by Dorigo, an Italian researcher, as a population-based heuristic algorithm with strong robustness [36]. It finds the optimal path by simulating the exploratory behavior of ants searching for food using distributed computing and pheromone updating mechanisms. The ant’s pathfinding starts from a certain point and chooses the next step according to a specific rule, then again chooses the next step according to a specific rule after reaching each next node until it reaches the goal. Pheromones are then left on the path traveled to guide subsequent ants; the higher the concentration of pheromones left behind, the greater the chance that the ants will choose that path.

The specific rules governing an ant’s choice of next step are called transition probabilities. For instance, an ant’s movement from node i to node j is determined by the transition probability $P_{ij}\left(t\right)$.(2)$$P_{ij}\left(t\right)=\left\{\begin{array}{cc} \frac{{\left[\tau_{ij}\left(t\right)\right]}^{\alpha }{\left[\eta_{ij}\left(t\right)\right]}^{\beta }}{\sum_{j\in allowed\left(k\right)}{\left[\tau_{ij}\left(t\right)\right]}^{\alpha }{\left[\eta_{ij}\left(t\right)\right]}^{\beta }} & j\in allowed\left(k\right)\\ 0 & j\notin allowed\left(k\right) \end{array}\right.$$

In Equation (2), $\tau_{ij}\left(t\right)$ denotes the pheromone concentration from node i to node j; $\eta_{ij}\left(t\right)$ represents the heuristic function; α is the importance factor for $\tau_{ij}\left(t\right)$; β is the heuristic expectation factor for $\eta_{ij}\left(t\right)$; $allowed\left(k\right)$ indicates the set of nodes that node i can visit next. Where $\eta_{ij}\left(t\right)$ is commonly expressed as:(3)$$\eta_{ij}\left(t\right)=\frac{1}{d_{ij}}$$

In Equation (3), $d_{ij}$ denotes the Euclidean distance from node i to node j.

When ants move, they release pheromones between the nodes they traverse. As ants traverse more paths, pheromone concentration increases, and it also evaporates with each iteration. After each iteration, pheromone concentration is updated using Equations (4)–(6).(4)$$\tau_{ij}\left(t+1\right)=\left(1-\rho \right)\tau_{ij}\left(t\right)+\Delta \tau_{ij}\left(t\right)$$(5)$$\Delta \tau_{ij}=\sum_{k=1}^{M}\Delta \tau_{ij}^{k}\left(t\right)$$(6)$$\tau_{ij}^{k}=\left\{\begin{array}{cc} \frac{Q}{L_{k}} & if\;ant\;k\;travels\;on\;node\;i\;to\;node\;j\\ 0 & otherwise \end{array}\right.$$
where ρ denotes the pheromone evaporation factor; $\Delta \tau_{ij}\left(t\right)$ denotes the total amount of pheromone released along the path from node i to node j; $L_{k}$ indicates the path length traversed by the *k-th* ant; Q is a constant.

### 2.3. Improvements to Ant Colony Optimization

#### 2.3.1. Initial Pheromone Distribution Rules

In the traditional ACO, the initial pheromone distribution is typically uniform and constant. Thus, the absence of differences between any two nodes results in blind exploration by ants during the initial phase and slows convergence speed. To address this uniform distribution and improve ants’ aimless exploration in the early stages, the initial pheromone concentration is determined by integrating node location information and obstacle information, as shown in Equation (7).(7)$$\tau_{ij}\left(0\right)=\frac{10d_{ST}}{d_{Si}+d_{iE}+b_{i}}$$
where $d_{Si}$ denotes the Euclidean distance from the current point to the starting node; $d_{iE}$ denotes the Euclidean distance from the current node to the target point; $d_{ST}$ denotes the Euclidean distance from the starting point to the target point; $b_{i}$ denotes obstacle information surrounding the current node, as shown in the following Equation (8).(8)$$b_{i}=\frac{\sum_{s=1}^{n}\sqrt{{\left(x_{s}-x_{i}\right)}^{2}+{\left(y_{s}-y_{i}\right)}^{2}}}{8}$$
where $\left(x_{i},y_{i}\right)$ denotes the coordinates of the current node; $\left(x_{s},y_{s}\right)$ denotes the coordinates of surrounding obstacles; n denotes the number of obstacles surrounding node i. The number 8 is a normalization constant used to convert the sum of obstacle distances into a standardized density metric. As shown in Equation (7), a larger value of $d_{Si}+d_{iE}$ results in lower pheromone concentration, reducing the likelihood of selecting that node. Simultaneously, higher obstacle information $b_{i}$ indicates denser surrounding obstacles, which also diminishes pheromone concentration. The initial pheromone distribution is illustrated in Figure 3, where color visually represents pheromone concentration variations. According to the initial pheromone distribution rules, pheromone concentration is higher near the line between the starting point S and the target point E, appearing as a deeper red hue. In areas with more obstacles, surrounding pheromones decrease, resulting in lower pheromone concentration and a bluer color.

#### 2.3.2. Improved State Transition Probability Rules

(1)Introducing universal gravitation as an inspiration function

In traditional ACO, the heuristic function used for transition probabilities only represents the reciprocal of the distance between the current node and each of the next available nodes. Nevertheless, since the distance between any two nodes in a grid map is always either 1 or $\sqrt{2}$, this approach suffers from blind search and lacks global optimization. To sovle this issue, the gravity search algorithm (GSA) [37] is introduced to replace the original heuristic function. It calculates the heuristic function by determining the gravitational pull between the current node and the next node.

The gravitational search algorithm is predicated on the mutual attraction between objects through universal gravitation. This force causes all objects to move toward those with greater mass.(9)$$F_{ij}^{d}\left(t\right)=G\left(t\right)\frac{M_{i}\left(t\right)M_{j}\left(t\right)}{R_{ij}\left(t\right)+\varepsilon }$$

In Equation (9), $F_{ij}^{d}\left(t\right)$ denotes the gravitational force exerted by object i on object j in the *d-th* dimension at time t; $G\left(t\right)$ denotes the gravitational constant at time t, which decreases over time; $M_{i}\left(t\right)$ and $M_{j}\left(t\right)$ denotes the gravitational masses of objects i and j, respectively; $R_{ij}\left(t\right)$ indicates the Euclidean distance between objects i and j; ε is a very small constant.(10)$$G\left(t\right)=G_{0}e^{-\sigma \frac{t}{T}}$$

In Equation (10), $G_{0}$ denotes; t denotes the current iteration step; T denotes the total number of iterations; σ is a constant.

The greater the mass M, the stronger the gravitational pull. The value of mass can be easily calculated using the corresponding fitness function.(11)$$fit_{i}\left(t\right)=\mathrm{norm}\left(p_{i}\left(t\right)-E\right)$$(12)$$M_{i}\left(t\right)=\frac{fit_{i}\left(t\right)}{\sum_{i=1}^{S}fit_{i}\left(t\right)}$$

In Equations (11) and (12), $fit_{i}\left(t\right)$ denotes the fitness value at position i at time t; $p_{i}\left(t\right)$ denotes the coordinates of the surrounding passable grid cells including the current position i; E denotes the target point coordinates; S is the total number of grid cells comprising the current position and its surrounding passable grid cells; norm denotes the Euclidean distance between two points.

(2)Angle-inspired function for introducing directional guidance

In most cases, path planning prioritizes smoother trajectories with fewer turns, making robotic motion more rational. To guide the robot toward optimal paths, a directional guidance angle heuristic is introduced into path planning. This enhances path smoothness and accelerates convergence, thereby improving robotic motion efficiency. The directional guidance angle heuristic function $O_{ij}\left(t\right)$ is expressed as shown in Equation (13).(13)$$O_{ij}\left(t\right)=\frac{1}{e^{\theta /2}}$$
where θ denotes the angle between the vector connecting the current node and the next node and the vector connecting the current node and the target node, which can be calculated using the following Equation (14).(14)$$\theta =\cos^{-1}\left(\frac{\left(x_{2}-x_{1}\right)\left(x_{3}-x_{2}\right)+\left(y_{2}-y_{1}\right)\left(y_{3}-y_{2}\right)}{\sqrt{{\left(x_{2}-x_{1}\right)}^{2}+{\left(y_{2}-y_{1}\right)}^{2}}\sqrt{{\left(x_{3}-x_{2}\right)}^{2}+{\left(y_{3}-y_{2}\right)}^{2}}}\right)$$

The coordinates $\left(x_{1},y_{1}\right)$, $\left(x_{2},y_{2}\right)$, and $\left(x_{3},y_{3}\right)$ denotes the current node, the next node, and the target node, respectively. A smaller value of θ denotes a smoother path that is closer to the shortest path, as shown in Figure 4.

Therefore, integrating the above rules yields the following new transition probability rules:(15)$$P_{ij}\left(t\right)=\left\{\begin{array}{cc} \frac{{\left[\tau_{ij}\left(t\right)\right]}^{\alpha }{\left[F_{ij}\left(t\right)\right]}^{\beta }{\left[O_{ij}\left(t\right)\right]}^{\gamma }}{\sum_{j\in allowde\left(k\right)}\;{\left[\tau_{ij}\left(t\right)\right]}^{\alpha }{\left[F_{ij}\left(t\right)\right]}^{\beta }{\left[O_{ij}\left(t\right)\right]}^{\gamma }} & j\in allowed\left(k\right)\\ 0 & j\notin allowed\left(k\right) \end{array}\right.$$

In Equation (15), γ is the heuristic factor for $O_{ij}\left(t\right)$.

#### 2.3.3. Improve the Pheromone Update Method

In ACO, the update method typically involves updating all paths uniformly, without prioritizing higher-quality paths. Over time, this approach tends to converge to local optima. To address this problem, paths are sorted by length and selected at a certain ratio. Superior paths receive additional rewards to increase their pheromone concentration, thereby accelerating convergence.

For pheromone updates, a proportion-based strategy is employed to modify pheromone concentrations along paths. Ants reaching the target point in each generation are evaluated, with paths ranking within the top $Z\left(N\right)\%$ receiving enhanced pheromone concentrations. The specific update formulas are shown in Equations (16)–(18):(16)$$\tau_{ij}\left(t+1\right)=\left(1-\rho \right)\tau_{ij}\left(t\right)+\Delta \tau_{ij}\left(t\right)+e_{1}\Delta \tau_{ij}^{bs}+e_{2}\frac{1}{L_{gb}}$$(17)$$\Delta \tau_{ij}^{bs}=\left\{\begin{array}{cc} 1/f_{m} & S_{f}\left(n\right)\geq S_{f}\left(\mathrm{ceil}\left(Z\left(N\right)Z\right)\right)\\ 0 & otherwise \end{array}\right.$$(18)$$Z\left(N\right)=\max\left(u-v\times N,Z_{\min}\right)$$
where $S_{f}$ denotes the paths sorted in descending order; N denotes the total number of paths reaching the target point in each generation, where n=1,2,3…N; $Z\left(N\right)$ is the dynamic threshold; u and v are constants, with u=0.4, v=0.002, and $Z_{min}=0.05$; $f_{m}$ is the average score of the top $Z\left(N\right)\%$ paths; $L_{gb}$ is the global shortest path; $e_{1}$ and $e_{2}$ are constants; ρ denotes the pheromone volatilization factor, calculated using Equation (19).(19)$$\rho =\frac{1}{1+e^{A\times K/k-B}}$$
where *K* denotes the maximum iteration times; *k* denotes the current iteration times; *A* and *B* are constants.

During the iteration process, the quantity of pheromones changes as ants explore. Early on, ants are encouraged to explore extensively. As exploration continues, pheromones begin to accumulate along high-quality paths. Later, search efforts are concentrated to conduct detailed searches near the optimal solution, accelerating convergence.

To prevent ants from clustering along a single path and avoid premature convergence to local optima that could cause the algorithm to stall, upper and lower bounds are set for pheromone concentration.(20)$$\tau_{ij}\left(t\right)=\left\{\begin{array}{cc} \tau_{\max}\left(t\right) & \tau_{ij}\left(t\right)\geq \tau_{\max}\left(t\right)\\ \tau_{ij}\left(t\right) & \tau_{\min}\left(t\right)<\tau_{ij}\left(t\right)<\tau_{\max}\left(t\right)\\ \tau_{\min}\left(t\right) & \tau_{ij}\left(t\right)\leq \tau_{\min}\left(t\right) \end{array}\right.$$(21)$$\left\{\begin{array}{c} \tau_{\max}\left(t\right)=\tau_{\max}\left(t-1\right)+\frac{1}{k}\left(\frac{23Q}{L_{gb}+0.625}-\tau_{\max}\left(t-1\right)\right)\\ \tau_{\min}\left(t\right)=\frac{\tau_{\max}\left(t-1\right)}{100} \end{array}\right.$$

In Equations (20) and (21), $\tau_{max}$ and $\tau_{min}$ denotes the upper and lower bounds of pheromone concentration, respectively; $\tau_{max}\left(t-1\right)$ denotes the maximum pheromone concentration of the previous generation; k denotes the current iteration count; and Q is a constant.

#### 2.3.4. Path Redundancy Inflection Point Optimization

Due to the inherent nature of grid maps, paths generated by algorithms are always diagonally connected. This typically results in paths with numerous turning points and insufficient smoothness. Therefore, this paper employs an optimization technique to reduce redundant turning points, aiming to shorten path lengths and enhance path smoothness. The optimized and original paths are shown in Figure 5.

The specific approach for optimizing the method is as follows:

Step 1: Identify the turning points along the original path and designate them as nodes to be optimized.

Step 2: Starting from the origin, sequentially search each subsequent turning point. Determine whether there are obstacles between the line segments connecting two turning points. If no obstacles exist, continue searching the next turning point. If obstacles are present, terminate the search and adopt the preceding turning point as the new node in the path.

Step 3: Using the newly obtained node as the starting point, repeat Step 2 until the endpoint is reached.

## 3. Fusion Path Planning Algorithm

### 3.1. Dynamic Window Approach for Local Path Planning

The DWA is a common local path planning algorithm. Its principle involves generating multiple predicted trajectories by selecting velocity combinations in the velocity sampling space, then scoring each trajectory. It determines the optimal trajectory and corresponding velocity combination for the robot at the current time step. The resulting velocity and angular velocity combinations are provided to the respective robot as control parameters. Due to limitations in global path planning capability, DWA is generally combined with global path planning algorithms. This combination enables the delivery robot to possess not only global path planning but also local dynamic obstacle avoidance capabilities.

#### 3.1.1. Establishment of a Robot Motion Model

The DWA algorithm primarily controls motion and determines position information through the robot’s linear velocity, angular velocity, and heading angle. Different kinematic models exist for different robot drive configurations. The robot motion model employed by delivery robots is a two-wheel differential motion model, as shown in Figure 6.

Assume the robot’s position at time t is ${\left[\begin{array}{ccc} x\left(t\right) & y\left(t\right) & \theta \left(t\right) \end{array}\right]}^{T}$. Over the time interval Δt, the position increment is given by Equation (22):(22)$$\left[\begin{array}{c} x\left(t+\Delta t\right)\\ y\left(t+\Delta t\right)\\ \theta \left(t+\Delta t\right) \end{array}\right]=\left[\begin{array}{c} x\left(t\right)\\ y\left(t\right)\\ \theta \left(t\right) \end{array}\right]+\left[\begin{array}{cc} \Delta t\cos\left(\theta_{t}\right) & 0\\ \Delta t\sin\left(\theta_{t}\right) & 0\\ 0 & \Delta t \end{array}\right]\left[\begin{array}{c} v_{t}\\ \omega_{t} \end{array}\right]$$
where $v_{t}$ denotes the robot’s velocity at time t, and $\omega_{t}$ denotes the robot’s angular velocity at time t.

#### 3.1.2. Establish a Speed Zone

After establishing the robot’s motion trajectory space, the velocity space must be defined. Within this defined velocity space, multiple velocity combinations are sampled at a specified resolution. Based on these sampled velocity combinations, multiple corresponding motion trajectories are simulated over a defined time period. These trajectories will be evaluated by the evaluation function to determine relatively optimal velocity combinations. The velocity space represents the robot’s speed range, constrained by factors associated with the delivery robot’s performance capabilities and environmental conditions, ultimately limiting it to a specific velocity range.

(1)Constrained by its maximum and minimum speeds, the delivery robot can reach a speed of $V_{m}$.


(23)
$$V_{m}=\left\{\left(v,\omega \right)|v\in \left[v_{\min},v_{\max}\right],\omega \in \left[\omega_{\min},\omega_{\max}\right]\right\}$$


In Equation (23), $v_{min}$ and $v_{max}$ denote the minimum and maximum linear velocities achievable by the delivery robot, respectively; $\omega_{min}$ and $\omega_{max}$ denote the minimum and maximum angular velocities achievable by the delivery robot, respectively.

(2)Constrained by motor performance, the delivery robot can achieve a speed of $V_{d}$.


(24)
$$V_{d}=\left\{\left(v,\omega \right)|v\in \left[v_{c}-{\dot{v}}_{b}\Delta t,v_{c}+{\dot{v}}_{a}\Delta t\right]\wedge \omega \in \left[\omega_{c}-{\dot{\omega }}_{b}\Delta t,\omega_{c}+{\dot{\omega }}_{a}\Delta t\right]\right\}$$


In Equation (24), $v_{c}$ and $\omega_{c}$ denote the linear velocity and angular velocity at the current time; ${\dot{v}}_{a}$ and ${\dot{v}}_{b}$ represent the maximum acceleration and maximum deceleration achievable in linear velocity under motor performance; ${\dot{\omega }}_{a}$ and ${\dot{\omega }}_{b}$ denote the maximum acceleration and maximum deceleration achievable in angular velocity under motor performance.

(3)Constrained by obstacles, the delivery robot can achieve a speed of $V_{a}$.


(25)
$$V_{a}=\left\{\left(v,\omega \right)|v\leq \sqrt{2\mathrm{dist}\left(v,\omega \right){\dot{v}}_{b}}\wedge \omega \leq \sqrt{2\mathrm{dist}\left(v,\omega \right){\dot{\omega }}_{b}}\right\}$$


In Equation (25), $\mathrm{dist}\left(v,\omega \right)$ denotes the distance between the simulation trajectory corresponding to $\left(v,\omega \right)$ and the nearest obstacle.

Taking the intersection of the above constraints yields the robot’s feasible velocity space, with the velocity range given by Equation (26):(26)$$V_{r}=V_{m}\cap V_{d}\cap V_{a}$$

#### 3.1.3. Optimization of the Evaluation Function

After sampling the velocity space, multiple predicted trajectories can be obtained based on the established robot motion model. At this point, these trajectories must be evaluated to select the optimal motion path, i.e., the optimal velocity combination. The evaluation function for the traditional DWA is shown in Equation (27):(27)$$G\left(v,\omega \right)=aHeading\left(v,\omega \right)+bDist\left(v,\omega \right)+cVel\left(v,\omega \right)$$
where $Heading\left(v,\omega \right)$ denotes the heading evaluation function, which assesses the angular difference between the predicted trajectory endpoint orientation and the target point at the current sampled velocity in velocity space; $Dist\left(v,\omega \right)$ is the distance evaluation function, representing the distance between the estimated trajectory endpoint and the nearest obstacle; $Vel\left(v,\omega \right)$ is the velocity evaluation function, representing the robot’s current linear velocity; a, b, and c are weighting coefficients.

To enhance the robot’s capability to avoid dynamic obstacles, dynamic obstacles are initially identified by comparing the obstacle’s position at the current time step with its position at the next time step. If the obstacle is determined to be dynamic, dynamic obstacle avoidance rules are applied. If the obstacle is static, static obstacle avoidance rules are applied. The distance evaluation subfunction is optimized within the evaluation function, specifically:(28)$$Dist\left(v,\omega \right)=\left\{\begin{array}{cc} JDist1\left(v,\omega \right)+KDist2\left(v,\omega \right) & d_{s}\neq 0\\ Dist1\left(v,\omega \right) & d_{s}=0 \end{array}\right.$$

In Equation (28), $Dist1\left(v,\omega \right)$ denotes the nearest distance between the endpoint of the predicted trajectory and static obstacles; $Dist2\left(v,\omega \right)$ denotes the nearest distance between the endpoint of the predicted trajectory and dynamic obstacles; $d_{s}$ represents the displacement of dynamic obstacles, as shown in Equation (29).(29)$$d_{s}=\sqrt{{\left(x_{o}\left(t\right)-x_{o}\left(t-1\right)\right)}^{2}+{\left(y_{o}\left(t\right)-y_{o}\left(t-1\right)\right)}^{2}}$$

When robots operate in areas with numerous obstacles, fluctuations in linear velocity become more frequent and pronounced, leading to reduced operational efficiency and unnecessary motor wear. Therefore, introducing a linear velocity variation sub-evaluation function into the assessment function, namely(30)$$Vel\_c\left(v,\omega \right)=e^{-w\left(v_{i+1}-v_{i}\right)}$$

In Equation (30), $Vel\_c\left(v,\omega \right)$ denotes the linear velocity change evaluation function; $v_{i}$ represents the linear velocity at the robot’s current position; $v_{i+1}$ denotes the linear velocity at the end of the predicted trajectory.

To ensure that the robot can better approach the global path and obtain a smoother path during its movement, a path proximity evaluation subfunction is introduced into the evaluation function, namely:(31)$$path\left(v,\omega \right)=\frac{abs\left(\left(x_{2}-x_{1}\right)\left(y_{p}-y_{1}\right)-\left(y_{2}-y_{1}\right)\left(x_{p}-x_{1}\right)\right)}{\sqrt{{\left(x_{2}-x_{1}\right)}^{2}+{\left(y_{2}-y_{1}\right)}^{2}}}$$

In Equation (31), $\left(x_{p},y_{p}\right)$ denotes the position at the end of the predicted trajectory, $\left(x_{1},y_{1}\right)$ represents the previous path key node, and $\left(x_{2},y_{2}\right)$ denotes the subsequent path key node.

Finally, the improved evaluation function is given by Equation (32):(32)$$\begin{array}{c} G\left(v,\omega \right)=aHeading\left(v,\omega \right)+bDist\left(v,\omega \right)+cVel\left(v,\omega \right)\\ +fVel\_c\left(v,\omega \right)+gPath\left(v,\omega \right) \end{array}$$
where f denotes the weight parameter for the linear velocity change assessment function; g denotes the weight parameter for the path proximity assessment function.

### 3.2. Dynamic Path Planning Algorithm Fusion

In traditional DWA, the problem often gets stuck in local optima. While IACO can achieve optimal solutions for global path planning in static environments, it lacks obstacle avoidance capabilities in complex environments with dynamic obstacles. To address this issue, the global path generated by the IACO is introduced as a guidance target for local dynamic planning.

By integrating the IACO with the enhanced DWA, the path turning points gained from the IACO serve as the required local sub-goal points for DWA. DWA performs segmented path planning using these sub-goal points to attain obstacle avoidance for the robot. The fusion algorithm enables globally optimal dynamic path planning and real-time obstacle avoidance. The path planning process of the fusion algorithm is illustrated in Figure 7.

The specific implementation steps are as follows:

Step 1: Create an environmental map by establishing the robot’s working environment using a grid map, and set the robot’s starting point and destination.

Step 2: Initialize relevant parameters such as ant population size, maximum iteration count, and various importance factors; initialize the initial pheromones on the map.

Step 3: Employ the IACO for global path planning to obtain a globally optimal path, and extract the sequence of key nodes along the path.

Step 4: Perform velocity sampling and generate predicted trajectories based on the robot’s kinematic model.

Step 5: According to the key nodes of the extracted path and the generated predicted trajectory, evaluate using the improved assessment function and select the optimal speed control as the output.

Step 6: Determine whether a local target point has been reached. If reached, select the next node as the new local target point and execute Step 4. If not reached, return to Step 4 to continue iteration.

Step 7: Determine whether the global target point has been reached. If the global target point has not been reached, return to step 6. If the global target point has been reached, terminate the loop.

## 4. Simulation Experiments and Results Analysis

In this section, all simulation experiments were conducted in a Windows 11 64-bit operating environment with a 3.2 GHz processor, 16 GB of RAM, and the simulation software was Matlab R2022b.

### 4.1. Determination of Key Parameters for the IACO

For IACO, different parameter combinations can significantly impact algorithm performance. However, no comprehensive theoretical method exists for determining optimal parameter configurations. In this paper, to identify key parameter combinations for IACO, we conducted multiple simulation experiments on primary parameters and performed statistical analysis.

To find the optimal parameter combination, the testing method involves altering one parameter at a time and all others are kept fixed. As shown in Equation (15), the primary parameters requiring testing are the pheromone importance factor *α*, the heuristic function expectation factor *β*, and the angular guidance heuristic factor *γ*. Under the condition of altering one parameter, four data metrics were analyzed: the optimal path length, the average path length at convergence, the average convergence iteration count, and the average iteration to reach the shortest path. Other parameters are listed in Table 1.

Simulation experiments were conducted on the grid map shown in Figure 8. The starting point was located at the blue grid in the upper left corner, and the target point was at the red grid in the lower right corner. To minimize the impact of errors, each parameter combination underwent 10 simulation runs.

For parameter *α*, its variation range is set to [1,8] with a step size of 0.5. Other parameters remain fixed: *β* = 2, *γ* = 5.5. Experimental results are shown in Figure 9. As seen in Figure 9a, the optimal path length consistently remains at 28.4132, indicating no significant variation in optimal path length regardless of *α* value. As shown in Figure 9b, the average path length at convergence exhibits an increasing trend as the value of *α* increases. Figure 9c,d reveal that *α* significantly influences the number of iterations. As *α* increases, both the average convergence iterations and the average iterations to reach the shortest path exhibit a pattern of first decreasing significantly and then increasing. The minimum average convergence iterations occur at *α* = 5.5. Therefore, the parameter value is determined as *α* = 5.5.

For parameter *β*, its variation range is set to [1,8] with a step size of 1. Other parameters remain fixed: *α* = 2, *γ* = 5.5. Experimental results are shown in Figure 10. Figure 10a indicates that the optimal path length consistently remains at 28.4132, demonstrating that the optimal path length shows no significant variation regardless of *β* value. Figure 10b reveals that when *β* ranges from 2 to 6, the average path length at convergence maintains a stable value, suggesting that *β* values within this range have no substantial impact on path length. Figure 10c,d reveal that as *β* increases, the average convergence iterations and the average iterations to reach the shortest path first decrease and then increase, with a minimum occurring at *β* = 4. Therefore, *β* = 4 is selected.

For parameter *γ*, its variation range is set to [1,10] with a step size of 0.5. Other parameters remain fixed: *α* = 2, *β* = 2. Experimental results are shown in Figure 11. Figure 11a indicates that when *γ* ranges from 1 to 8, the optimal path length remains constant at 28.4132. However, *γ* exhibits significant changes beyond 8. As shown in Figure 11b, the average path length at convergence first decreases and then increases as *γ* increases. The average path length remains relatively stable within the range of *γ* = 2.5 to 5.5. Figure 11c,d demonstrate that *γ* significantly affects the average convergence iterations and the average iterations to reach the shortest path, with a minimum value occurring at *γ* = 7.5. However, at *γ* = 7.5, the average path length at convergence falls outside the optimal range, making it prone to local optima. Finally, through experimental validation of *γ* within the range of 2.5 to 5.5, *γ* = 4.5 was ultimately selected to achieve better performance.

In summary, after statistical experimentation, the optimal configuration of parameter for the IACO was determined as *α* = 5.5, *β* = 4, and *γ* = 4.5. Substituting this parameter set into the algorithm, its impact on path planning is illustrated in Figure 12. The optimal path generated by the IACO achieved a length of 28.4132, with 6 turns, and 4 iterations.

### 4.2. Comparative Simulation Experiments of the IACO

In this section, multiple groups of experiments were conducted to better verify the performance of the IACO in path planning. These experiments involved various maps and were performed across multiple distinct experimental environments to validate the efficiency and adaptability of the IACO. Furthermore, the IACO was compared with other algorithms to demonstrate its effectiveness.

#### 4.2.1. Performance Validation in Different Complex Environments

Four distinct environmental maps were designed to test the adaptability of the proposed IACO. This set of experiments compared the proposed algorithm with the traditional ACO on different maps. Four distinct maps are illustrated in Table 2 and Figure 13. On the map, the blue grid serves as the starting point, while the red grid serves as the destination. In the algorithm parameter configuration, the proposed IACO shares the same common parameter settings as the traditional ACO. Specifically, the traditional ACO parameters are *M* = 50, *K* = 100, *α* = 5.5, *β* = 4, *ρ* = 0.3, *Q* = 1. The proposed IACO parameters are set as *M* = 50, *K* = 100, *α* = 5.5, *β* = 4, *γ* = 4.5. Other parameters are shown in Table 1. To minimize the impact of random errors, each algorithm was run independently 20 times. The experimental results are shown in Figure 13 and Table 3. Figure 13 displays the optimal paths generated by each algorithm. Additionally, Table 3 presents three evaluation metrics: the percentage improvement in optimal path length (Length improve Best), the percentage improvement in average path length (Length improve Mean), and the percentage improvement in optimal path turning points (Turn improve Best).

As shown in Figure 13 and Table 3, the IACO demonstrates excellent adaptability across various environments. Across four distinct map characteristics, the improved algorithm outperforms the traditional ACO. It exhibits significantly superior performance in terms of fewer turns required. Across the four maps, the improved algorithm reduced the number of turns by 25%, 41.67%, 25%, and 71.43%, respectively, compared to the traditional ACO. With respect to path length performance metrics, the IACO demonstrated significant advantages in optimal path length, average path length, and path length standard deviation. The IACO achieved optimal path lengths of 27.218, 29.458, 28.951, and 33.066 across four maps, all of which were shorter than those obtained by the traditional ACO. The average path lengths achieved by the IACO across the four maps were 27.218, 29.521, 29.206, and 33.416, respectively, all lower than those of the traditional ACO. Notably, in Map 4, the average path length improvement reached 30.03%. The standard deviation of paths obtained across the four maps is relatively small, with Map 1 exhibiting a standard deviation of zero, indicating that the proposed IACO demonstrates high stability. Experimental results demonstrate that the IACO possesses advantages and adaptability in diverse environmental conditions.

#### 4.2.2. Comparative Experiments with Other Algorithms

(1)Comparative experiment 1.

In Comparative experiment 1, the grid map used in Reference [17] served as the experimental environment. The proposed IACO algorithm was compared against six other intelligent algorithms: the improved ACO algorithms (IACO and APACA) mentioned in Reference [38], the IHMACO algorithm proposed in Reference [17], the A*, the Dijkstra, and the best-first-search. To minimize errors, all experiments were independently executed 20 times. The obtained optimal path trajectories and convergence curves are shown in Figure 14 and Figure 15, respectively, while the statistical data are presented in Table 4.

The data in Figure 14 and Figure 15 and Table 4 demonstrate that the proposed IACO is superior to the other six algorithms in terms of path length, turn times, and convergence speed. Regarding path length, the proposed IACO achieves the minimum values for both optimal and average path lengths, both at 27.63. Additionally, the standard deviation is zero, indicating algorithmic stability. Except for matching the number of turns with IHMACO, the proposed IACO exhibits the fewest turns at 4, indicating superior path smoothness. Despite sharing the same number of turns as IHMACO, the improved algorithm achieves shorter paths. Furthermore, Figure 15 demonstrates that the improved algorithm converges faster. Compared to the IACO in the literature [38], APACA, IHMACO, A*, Dijkstra, and best-first-search algorithms, the proposed IACO demonstrates superior performance advantages. This demonstrates that the proposed improvement mechanism is effective and exhibits superior overall performance.

(2)Comparative experiment 2.

In Comparative experiment 2, the proposed IACO was compared with five other intelligent algorithms, including the traditional ACO, the ZACO mentioned in [39], the MAACO proposed in [12], the A*, and the Dijkstra. Experiments were performed on the 20 × 20 map model described in Reference [12], as shown in Figure 15. To minimize error, all experiments were independently executed 20 times. The obtained optimal path trajectories and convergence curves are illustrated in Figure 16 and Figure 17, respectively, with the corresponding statistical data provided in Table 5.

The data in Figure 16 and Figure 17 and Table 5 show that, except for the Dijkstra algorithm, the other four algorithms all converge stably to 29.21. The IACO proposed in this paper, by adopting the strategy of redundant turning point optimization, further reduces the path length to 27.88. It also further reduces the number of turning points to 5. Figure 17 and Table 5 demonstrate that while ensuring optimal solutions, the algorithm achieves a faster convergence rate with an average of 4.7 iterations—the swiftest among all methods. This acceleration stems from the incorporation of an initial pheromone strategy and an enhanced transition mechanism. In summary, the experimental results indicate that the IACO attains shorter path lengths and faster convergence rates compared to traditional ACO, improved ACO in [39], and MAACO algorithms. Furthermore, it yields superior solutions relative to the A* and Dijkstra algorithms.

(3)Comparative experiment 3.

In experiment 3, to further verify the performance of the IACO, performance tests were conducted in a more complex environment. The 30 × 30 map model described in [12] was employed for testing. The IACO was compared against the traditional ACO, the MRCACO proposed in [40], the AIACSE proposed in [13], the MAACO proposed in [12], the A* algorithm, and the Dijkstra algorithm. To minimize error, all experiments were independently executed 20 times. The obtained optimal path trajectories are shown in Figure 18, and the statistical data are presented in Table 6.

Based on the data in Table 6, despite larger and more complex maps, the IACO algorithm still demonstrates superior performance. Regarding path length, the IACO algorithm achieved the shortest optimal path length of 41.4635. Additionally, the standard deviation of path length was zero, indicating both the ability to find optimal paths and excellent stability. Algorithms such as MRCACO, MAACO, A*, and Dijkstra also achieve relatively optimal paths with good stability. However, the paths they find contain a higher number of turning points. The IACO, on the other hand, achieves fewer turning points, with only 4 turns. This demonstrates that the strategies of redundant turning point optimization, improved transition mechanism, and enhanced pheromone update strategy effectively enhance the algorithm’s performance, leading to superior path planning.

### 4.3. Fusion Algorithm Simulation Experiments

In the hybrid algorithm simulation experiment, the kinematic parameters of the delivery robot are shown in Table 7. The evaluation parameters for the DWA algorithm are: a = 0.3, b = 0.12, c = 0.2, f = 0.15, g = 0.2, J = 0.35, K = 0.65. The prediction period is 3 s. In the following experiment, the starting point is the grid cell marked by the blue triangle, the endpoint is the grid cell marked by the red circle, the solid blue line represents the trajectory of the fusion algorithm, and the dashed black line denotes the global path planned by the ACO.

#### 4.3.1. Experimental Evaluation and Analysis of Fusion Algorithm Effectiveness

To evaluate the effectiveness of the fusion algorithm, comparisons were conducted on the same map between the traditional DWA and the fusion algorithm incorporating global path planning. Figure 19 illustrates the paths generated by the traditional DWA algorithm and the improved fusion algorithm, while Table 8 presents the statistical data from the comparative experiments.

#### 4.3.2. Experimental Analysis of Fusion Algorithm Obstacle Avoidance

Taking the actual layout diagram of a cafeteria as an example, as shown in Figure 20a, a fusion path planning algorithm is applied to perform a global search for the robot’s walking path, identifying an optimal route that is relatively short, smooth, and highly secure. Obstacles in the space are arranged according to patterns based on the actual restaurant layout (height is not considered). During robot operation, the structure, volume, and position information of obstacles remain fixed and known. Based on this, the robot’s workspace model defines a simulated grid map according to the actual restaurant scale.

Determine the grid map size based on the delivery robot’s dimensions and the actual environment. Assuming the robot’s projected footprint on the floor is 0.5 m × 0.5 m and the restaurant area is 20 m × 20 m, the grid size is set slightly larger than the robot’s maximum diameter to allow free movement within the grid. Therefore, a grid map size of 20 × 20 is adopted, as shown in Figure 20b.

In a 20 × 20 map of a cafeteria dining hall, the feasibility of the fusion algorithm is validated by setting static and dynamic obstacles. In the experimental map, two static obstacles are placed, as shown by the gray circles in Figure 21. Additionally, two dynamic obstacles are placed, as shown by the orange circles in Figure 22.

In Figure 21 and Figure 22, The black circle denotes the path key point. The path planning data results are shown in the diagram Table 9 and Figure 23. Table 9 displays the path length, planning time, number of control nodes, and minimum distance from obstacles during navigation achieved by the fusion algorithm throughout the path planning process. Figure 23 illustrates the variation in speed during obstacle avoidance and the path deviation relative to the distance key point.

(1)Static obstacle avoidance experiment analysis.

Figure 21 illustrates the obstacle avoidance process of the fusion algorithm in a map containing static obstacles. Before encountering obstacles, the robot effectively navigates along key nodes specified by the global path. When static obstacles appear ahead, the robot detects them and executes avoidance maneuvers. As shown in Figure 21, the robot successfully bypasses both obstacles on straight paths and those at corners. After obstacle avoidance, it resumes following key nodes along the global path until reaching the destination. The path results are shown in Table 9. The final planned path length is 28.452 m, with 766 control nodes and a simulation time of 386.98 s. During navigation, the minimum distance to obstacles is 0.2 m, occurring between control nodes 0 and 30, at which point the robot operates at a slower speed.

(2)Dynamic obstacle avoidance experiment analysis.

Figure 22 illustrates the obstacle avoidance process of the fusion algorithm in a map containing dynamic obstacles. During the movement, the robot dynamically detects obstacles, determining whether they are moving. For moving obstacles, it follows dynamic obstacle avoidance rules; for stationary obstacles, it follows static obstacle avoidance rules. As shown in Figure 22, the robot successfully navigated around both dynamic and static obstacles at two locations, respectively, and safely reached the destination. The path results are shown in Table 9. The final planned path length is 28.94 m, with 800 control nodes and a simulation time of 419.25 s. During navigation, the minimum distance to obstacles is 0.2 m, occurring between control nodes 0 and 30 as in Experiment (1), at which point the robot operates at a slower speed.

Figure 23a illustrates the variation in linear velocity of the robot during the simulation experiment. Linear velocity 1 represents the linear velocity variation in Experiment (1), while linear velocity 2 denotes the linear velocity variation in Experiment (2). As can be seen from the diagram, the linear velocity exhibits relatively stable variation, enabling timely deceleration upon encountering obstacles and navigating around them at a comparatively safe speed. Figure 23b illustrates the path deviation during the simulation experiment. Error1 denotes the degree of path deviation in Experiment (1), while Error2 denotes the degree of path deviation in Experiment (2). In Experiment (1), the maximum deviation of the path distance from the target point was 0.29 m, with an average deviation of 0.188 m. In Experiment (2), the maximum deviation of the path distance from the target point was 0.74 m, with an average deviation of 0.318 m. This indicates that the robot must deviate from the target point to avoid obstacles, overall continues to travel along the global path.

## 5. Conclusions

Robot navigation relies heavily on path planning, hence studying path planning algorithms is necessary. This paper overcomes the shortcomings and limitations of traditional ACO and DWA algorithms by optimizing both approaches. It then integrates the IACO and DWA algorithms to improve the overall performance and obstacle avoidance capabilities of the path planning system.

To address the limitations of traditional ACO, the following improvement strategies are proposed: A strategy for initial pheromone distribution based on node positions and obstacle information is introduced. A gravitational mechanism and directional angle heuristic function are adopted as new heuristic information for state transition rules. A new pheromone update rule is implemented that sorts paths by length and adds extra pheromone to paths with optimal lengths. By reducing redundant turning points in global paths, the algorithm optimizes the issue of excessive path turns in traditional ACO, resulting in smoother paths.

To optimize the local path planning performance of the DWA, this study enhances its evaluation function to better track the global path while simultaneously improving dynamic obstacle avoidance capabilities. To this end, three evaluation subfunctions have been introduced based on the evaluation function: the dynamic obstacle distance evaluation subfunction, the linear velocity change evaluation subfunction, and the path proximity evaluation subfunction.

Simulation experiments demonstrate that compared to the traditional ACO across four environments, the IACO decreased the average path length by 13.98%, 17.53%, 20.32%, and 30.03%, respectively. It also decreased the number of path turns by 25%, 41.67%, 25%, and 71.43%, respectively. Comparisons with several other intelligent algorithms further demonstrated that the IACO exhibits superior advantages in path length, smoothness, and convergence iterations. Finally, the integration of the IACO and DWA algorithms enabled the fusion algorithm to effectively plan optimal paths and execute superior obstacle avoidance strategies in both static and dynamic obstacle avoidance experiments.

## Figures and Tables

**Figure 1 sensors-26-00072-f001:**
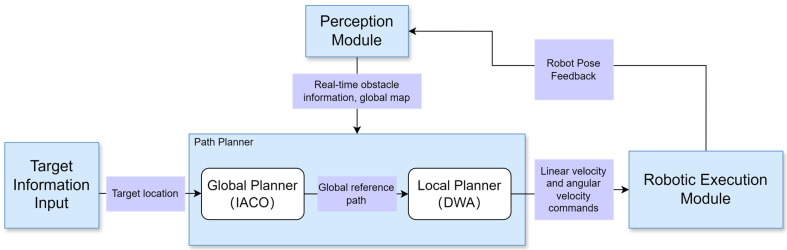
Path Planning and Fusion Control Framework.

**Figure 2 sensors-26-00072-f002:**
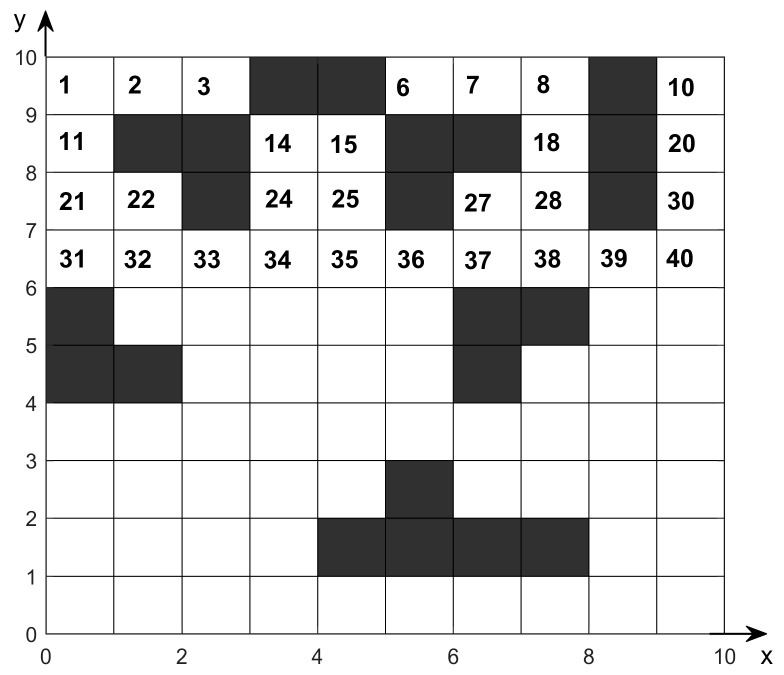
Grid map environment model.

**Figure 3 sensors-26-00072-f003:**
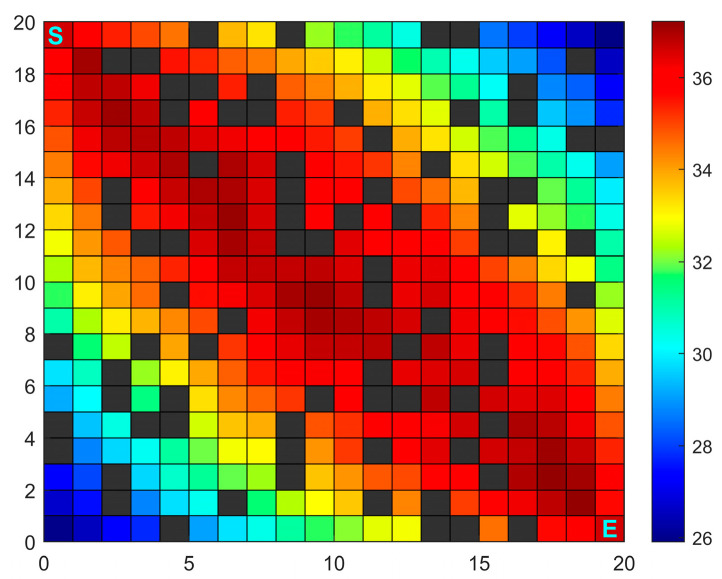
Schematic diagram of initial pheromone concentration distribution.

**Figure 4 sensors-26-00072-f004:**
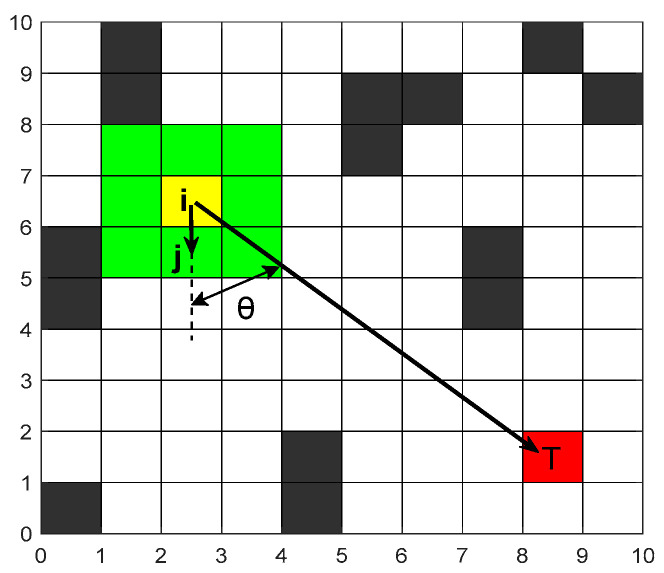
Angle diagram.

**Figure 5 sensors-26-00072-f005:**
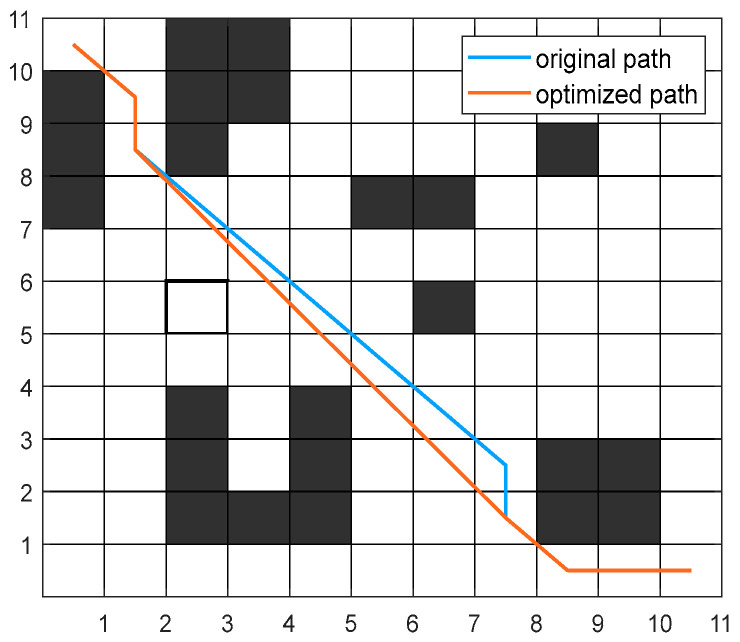
Path turning point optimization diagram.

**Figure 6 sensors-26-00072-f006:**
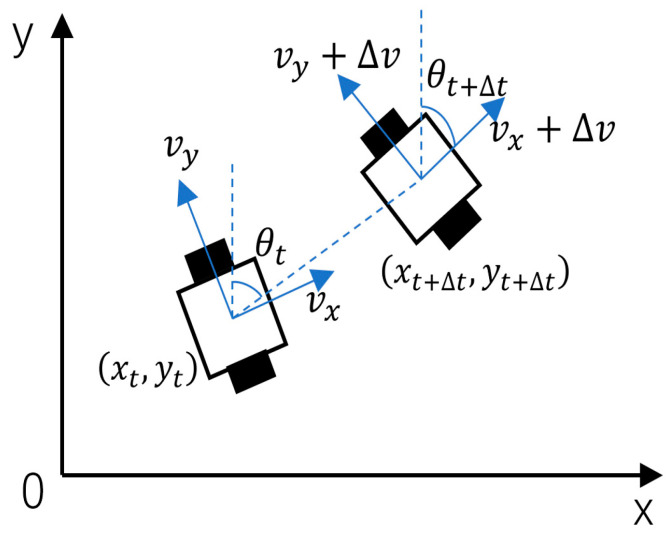
Delivery robot kinematic model.

**Figure 7 sensors-26-00072-f007:**
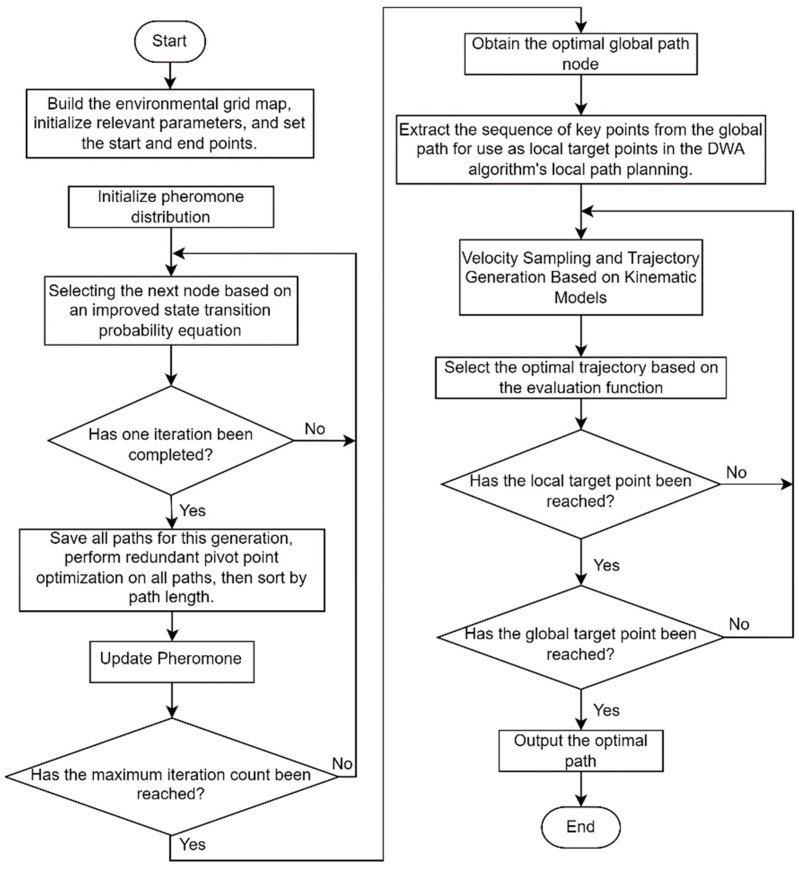
Fusion algorithm flowchart.

**Figure 8 sensors-26-00072-f008:**
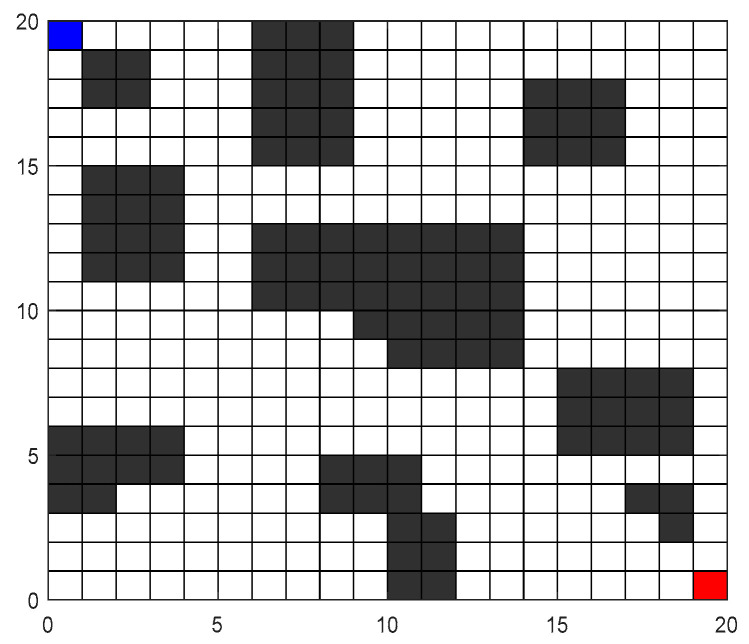
Parameter-optimized environment map.

**Figure 9 sensors-26-00072-f009:**
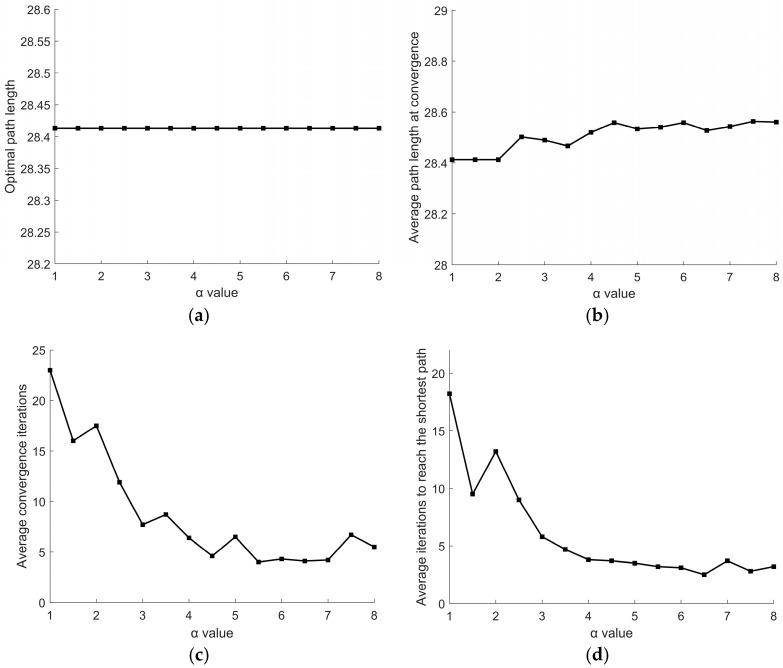
Effect of different *α* values on path: (**a**) optimal path length; (**b**) average path length at convergence; (**c**) average convergence iterations; (**d**) average iterations to reach the shortest path.

**Figure 10 sensors-26-00072-f010:**
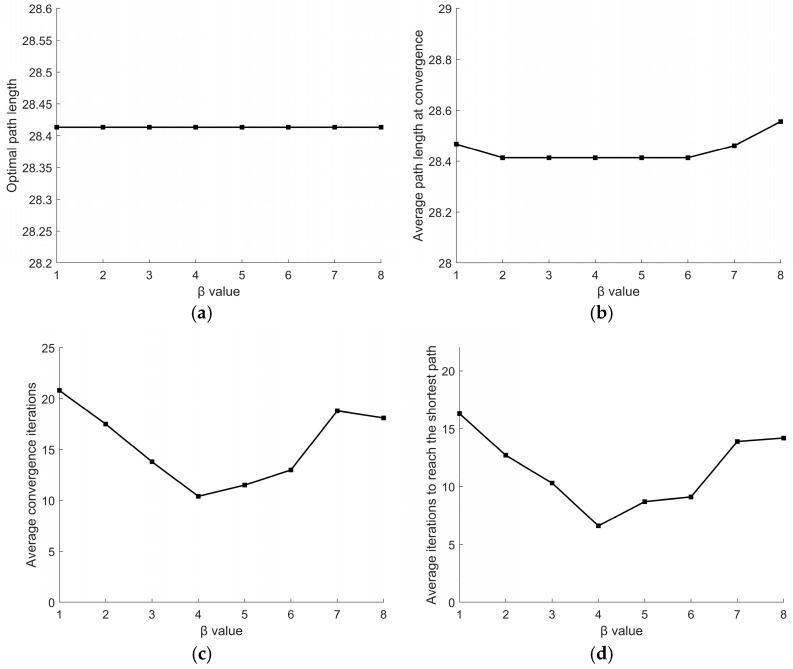
Effect of different *β* values on the path: (**a**) optimal path length; (**b**) average path length at convergence; (**c**) average convergence iterations; (**d**) average iterations to reach the shortest path.

**Figure 11 sensors-26-00072-f011:**
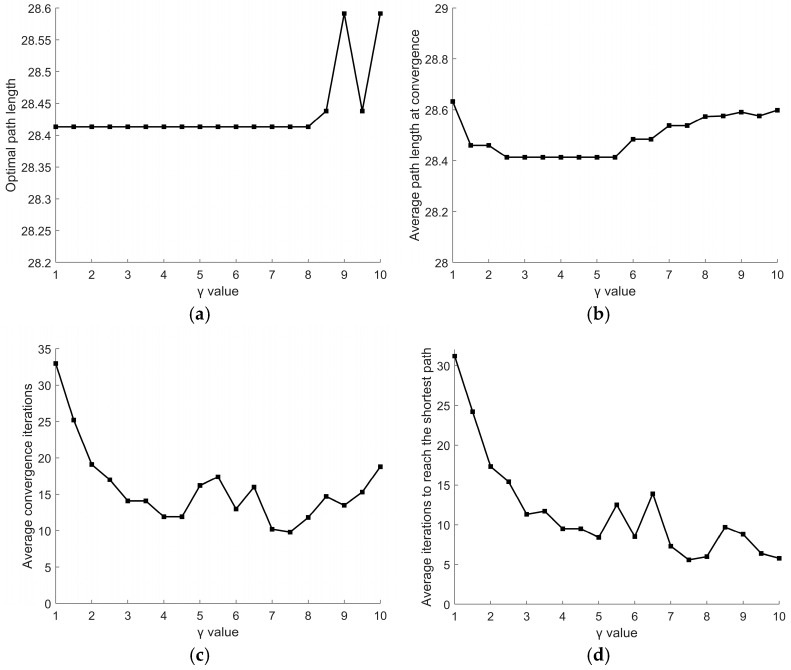
Effect of different *γ* values on path: (**a**) optimal path length; (**b**) average path length at convergence; (**c**) average convergence iterations; (**d**) average iterations to reach the shortest path.

**Figure 12 sensors-26-00072-f012:**
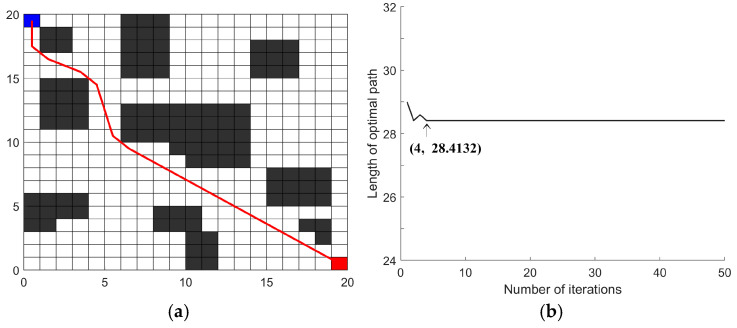
Results validation: (**a**) Optimal path; (**b**) convergence curve.

**Figure 13 sensors-26-00072-f013:**
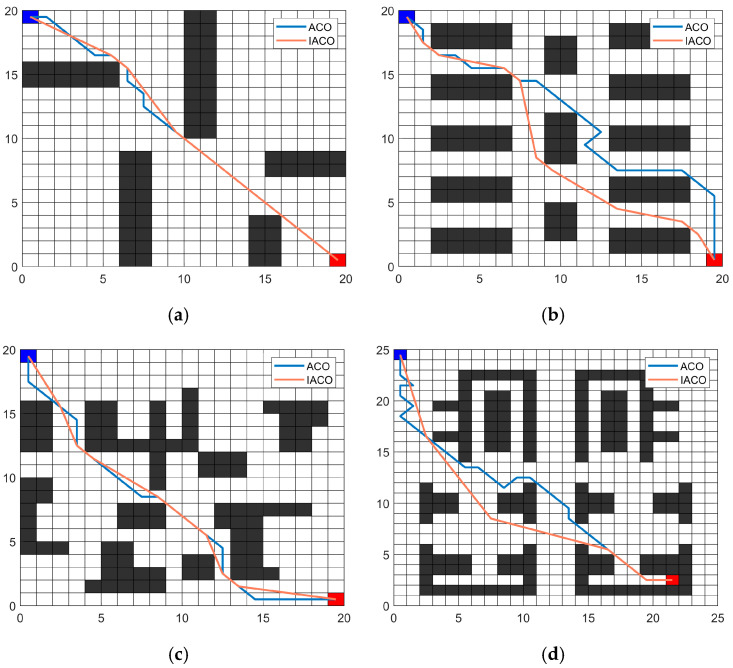
Optimal path across four types of maps: (**a**) Map 1; (**b**) Map 2; (**c**) Map 3; (**d**) Map 4.

**Figure 14 sensors-26-00072-f014:**
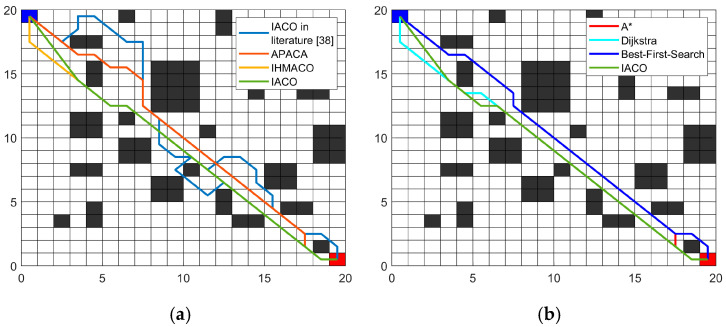
Optimal paths: (**a**) IACO in the literature [38], APACA, IHMACO, and IACO; (**b**) A*, Dijkstra, Best-First-Search, and IACO.

**Figure 15 sensors-26-00072-f015:**
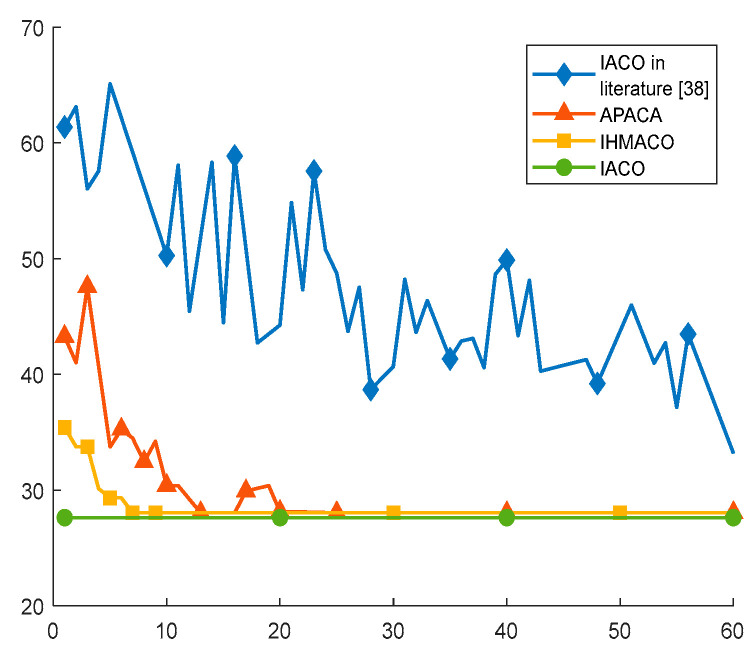
Convergence iteration curves for IACO in the literature [38], APACA, IHMACO, and IACO.

**Figure 16 sensors-26-00072-f016:**
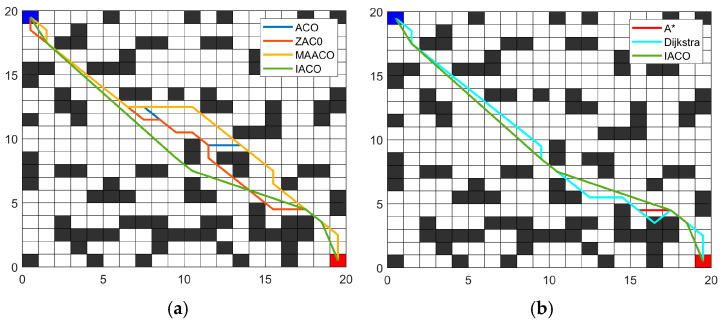
Optimal paths: (**a**) ACO, ZACO, MAACO and IACO; (**b**) A*, Dijkstra, and IACO.

**Figure 17 sensors-26-00072-f017:**
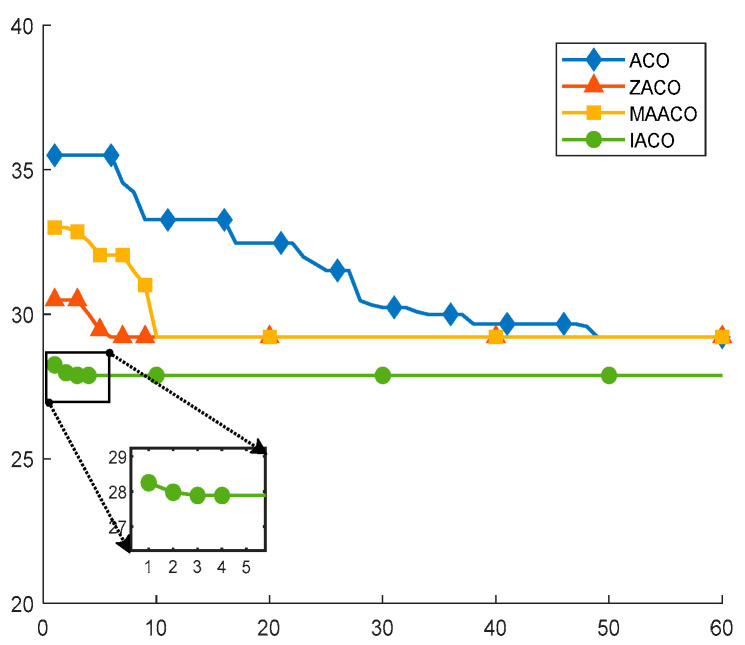
Convergence iteration curves for ACO, ZACO, MAACO, and IACO.

**Figure 18 sensors-26-00072-f018:**
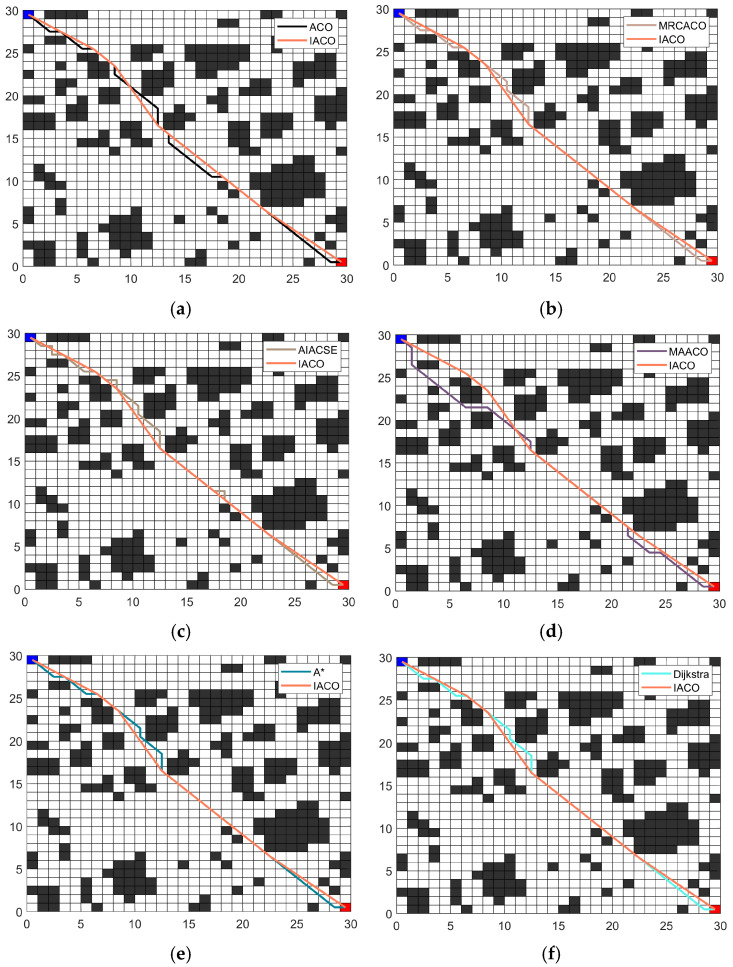
Optimal paths: (**a**) ACO; (**b**) MRCACO; (**c**) AIACSE; (**d**) MAACO; (**e**) A*; (**f**) Dijkstra.

**Figure 19 sensors-26-00072-f019:**
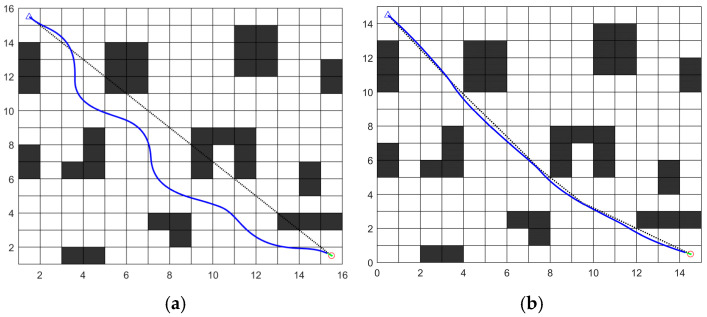
Path planning results comparison: (**a**) Traditional DWA; (**b**) Improved fusion algorithm.

**Figure 20 sensors-26-00072-f020:**
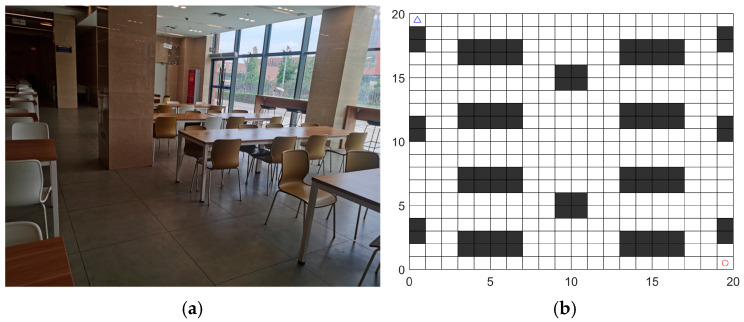
Experimental environment in the cafeteria: (**a**) Actual layout of the cafeteria dining area (partial view); (**b**) cafeteria grid map (20 × 20).

**Figure 21 sensors-26-00072-f021:**
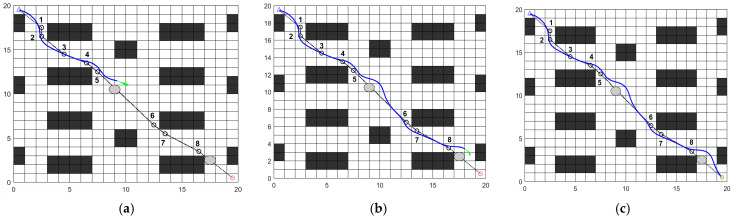
Static obstacle avoidance process: (**a**) Avoid the first static obstacle; (**b**) Avoid the second static obstacle; (**c**) Reaching the destination.

**Figure 22 sensors-26-00072-f022:**
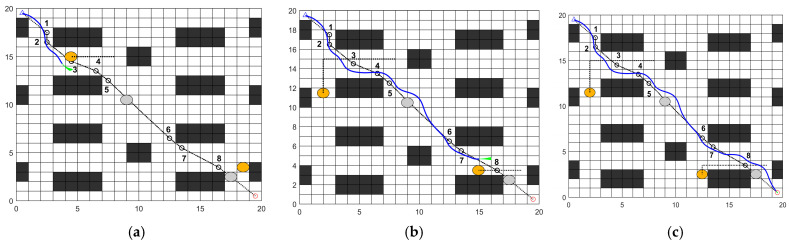
Dynamic obstacle avoidance process: (**a**) Avoid the first dynamic obstacle; (**b**) Avoid the second dynamic obstacle; (**c**) Reaching the destination.

**Figure 23 sensors-26-00072-f023:**
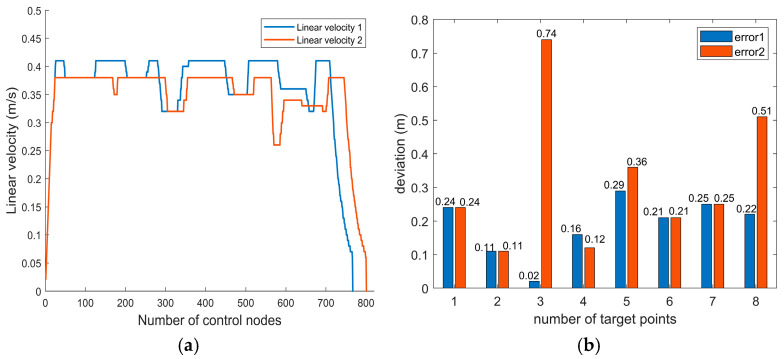
Speed Variations and Path Deviations During Obstacle Avoidance: (**a**) Linear Speed Output Results; (**b**) Path Distance Deviation.

**Table 1 sensors-26-00072-t001:** Relevant parameters involved in the IACO.

**Parameters**	*K*	*M*	*G* _0_	*σ*	*A*	*B*	*e* _1_	*e* _2_	*Q*
**Value**	100	50	10	2	0.5	0.05	0.5	200	100

**Table 2 sensors-26-00072-t002:** Characteristics and design objectives of different complex environments.

Map	Features	Design Objectives
1	Simple rectangular obstacle	Testing the capability to handle simple obstacles
2	A large number of rectangular obstacles distributed uniformly and discretely	Testing the for handling multi-channel and multi-turn scenarios
3	Obstacles of varying sizes and shapes that are randomly distributed	Testing the capability to handle obstacles of varying sizes and shapes that are randomly distributed
4	Irregularly shaped complex obstacles with traps	Testing the IACO ability to handle traps and complex maps

**Table 3 sensors-26-00072-t003:** Performance validation results of traditional ACO and IACO on four types of maps.

Map	Algorithm	Path Length			Turn Times	Length Improve		Turn Improve
		Optimal	Mean	Std.	Best	Best	Mean	Best
1	ACO	28.042	31.642	2.583	4	2.94%	13.98%	25%
	IACO	27.218	27.218	0	3			
2	ACO	32.385	35.797	1.704	12	9.04%	17.53%	41.67%
	IACO	29.458	29.521	0.130	7			
3	ACO	30.385	36.655	3.502	8	4.72%	20.32%	25%
	IACO	28.951	29.206	0.223	6			
4	ACO	37.284	47.759	3.861	14	11.31%	30.03%	71.43%
	IACO	33.066	33.416	0.344	4			

**Table 4 sensors-26-00072-t004:** Comparative results of experiment 1.

Algorithm	Path Length			Turn Times	Length Improve		Turn Improve
	Optimal	Mean	Std.	Best	Best	Mean	Best
IACO in the literature [38]	30.380	35.357	4.3764	24	9.05%	21.85%	83.33%
APACA	28.038	29.184	1.0020	9	1.46%	5.32%	55.56%
IHMACO	28.038	28.038	0	4	1.46%	1.46%	0
A*	28.038	28.038	0	7	1.46%	1.46%	42.86%
Dijkstra	28.038	28.038	0	7	1.46%	1.46%	42.86%
Best-First-Search	28.038	28.038	0	7	1.46%	1.46%	42.86%
IACO	27.630	27.630	0	4	-	-	-

**Table 5 sensors-26-00072-t005:** Comparative results of comparative experiment 2.

Algorithm	Path Length			Iteration			Turn Times	Path Improve		Turn Improve
	Optimal	Mean	Std.	Best	Mean	Std.	Best	Best	Mean	Best
ACO	29.21	33.55	1.58	49	54.35	2.26	11	4.54%	16.89%	54.55%
ZACO	29.21	-	-	5	-	-	10	4.54%	-	50%
MAACO	29.21	29.21	0	9	9.8	0.72	7	4.54%	4.54%	28.57%
A*	29.21	29.21	0	-	-	-	9	4.54%	4.54%	44.44%
Dijkstra	30.04	30.04	0	-	-	-	9	7.18%	7.18%	44.44%
IACO	27.88	27.88	27.88	3	4.7	1.1	5	-	-	-

**Table 6 sensors-26-00072-t006:** Comparative results of experiment 3.

Algorithm	Path Length			Turn Times	Path Improve		Turn Improve
	Optimal	Mean	Std.	Best	Best	Mean	Best
ACO	43.3553	47.9413	1.3199	13	4.36%	13.51%	69.23%
MRCACO	42.7696	42.7696	0	9	3.05%	3.05%	55.56%
AIACCSE	44.5269	44.5269	0	17	6.88%	6.88%	76.47%
MAACO	42.7696	42.7696	0	7	3.05%	3.05%	42.86%
A*	42.7696	42.7696	0	9	3.05%	3.05%	55.56%
Dijkstra	42.7696	42.7696	0	9	3.05%	3.05%	55.56%
IACO	41.4635	41.4635	0	4	-	-	-

**Table 7 sensors-26-00072-t007:** Kinematics Parameters of Delivery Robots.

Parameters	Value
Maximum linear velocity (m/s)	1
Maximum angular velocity ((°)/s)	20
Acceleration (m/s^2^)	0.2
Angular acceleration ((°)/s^2^)	50
Linear velocity resolution (m/s)	0.01
Angular velocity resolution (°/s)	1

**Table 8 sensors-26-00072-t008:** Experimental data for the traditional DWA algorithm and the improved fusion algorithm.

	Path Length (m)	Number of Control Nodes	Times (s)
Traditional DWA	21.362	487	188.2463
Improved fusion algorithm	19.940	479	185.5991

**Table 9 sensors-26-00072-t009:** Obstacle Avoidance Experiment Statistics.

	Path Length (m)	Number of Control Nodes	Time (s)	Minimum Distance (m)
Experiment (1)	28.452	766	386.98	0.2
Experiment (2)	28.940	800	419.25	0.2

## Data Availability

The original contributions presented in this study are included in the article. Further inquiries can be directed to the corresponding author.

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
