# Peer review of "Path Planning for Delivery Robots Based on an Improved Ant Colony Optimization Algorithm Combined with Dynamic Window Approach"

_sensors, 2025, doi:10.3390/s26010072_

Round 1
Reviewer 1 Report
Comments and Suggestions for Authors
The paper presents a hybrid path planning approach for delivery robots by combining an improved ant colony optimization (IACO) with the dynamic window approach (DWA), addresses relevant challenges in the field. Overall, the manuscript is well-written, and the contributions are potentially valuable for the community. However, several significant issues must be addressed to improve the clarity, rigor, and contribution of the paper. The following comments are provided for the authors' consideration:
- The overall formatting of the manuscript is poor, which greatly hinders readability and makes some content difficult to identify accurately.
- In the "Introduction" section, although the domestic and international research status is introduced, it does not objectively analyze the shortcomings or limitations of existing studies. This omission makes it difficult to fully demonstrate the necessity and innovativeness of the present work.
- In Equation (8) for the obstacle information term b_i, (y_s + y_i) should be (y_s - y_i) (a consistent difference operation is required for Euclidean distance calculation), and the rationale for dividing by 8 is not explained (Section 2.3.1, Page 7, Line 230). The authors need to correct the formula and provide a clear justification for the division factor.
- In the explanation of Figure 2: "According to the initial pheromone distribution rules, pheromone concentration is higher near the line between the starting point and the target point, with a brighter yellow color distribution," the specific positions of the starting point and target point are not clearly marked in the figure. Additionally, the statement "In areas with more obstacles, surrounding pheromones decrease, resulting in lower pheromone concentration and a duller color" is difficult to distinguish clearly due to the indistinct color gradient for regions with varying obstacle densities. The authors should add labels for the start and target points in Figure 2 and enhance the color contrast or provide quantitative annotations to clarify pheromone concentration differences.
- A critical omission is the lack of performance comparison between the fused algorithm and standalone IACO or DWA. The synergistic benefits of the fusion—such as improvements in path length, real-time performance, or obstacle avoidance success rate—are not analyzed.
- The real-world experimental results are only qualitatively described (e.g., "successful avoidance", "safely reached"). Quantitative metrics—such as path deviation during avoidance, linear velocity fluctuation, response time, path length, and travel time—are missing. Additionally, the paths in Figures 18 and 19 appear to pass close to obstacles, suggesting potential safety risks. The authors should discuss this behavior and supplement the results with quantitative safety measures, such as the minimum distance to obstacles during navigation.
Author Response
Comments 1 :The overall formatting of the manuscript is poor, which greatly hinders readability and makes some content difficult to identify accurately.
Response 1:Thank you for pointing this out. We agree with this comment. Therefore, we have already revised the overall typesetting of the manuscript, including images and equations.
Comments 2 :In the "Introduction" section, although the domestic and international research status is introduced, it does not objectively analyze the shortcomings or limitations of existing studies. This omission makes it difficult to fully demonstrate the necessity and innovativeness of the present work.
Response 2:Agreed. Therefore, we have incorporated a discussion of the shortcomings or limitations of prior research within the introduction section. The revised contents are located on page 4 of the manuscript, lines 159 to 164 inclusive.
Comments 3:In Equation (8) for the obstacle information term b_i, (y_s + y_i) should be (y_s - y_i) (a consistent difference operation is required for Euclidean distance calculation), and the rationale for dividing by 8 is not explained (Section 2.3.1, Page 7, Line 230). The authors need to correct the formula and provide a clear justification for the division factor.
Response 3:We are most grateful for your identification of the error in the manuscript. Consequently, we have amended equation (8) (page 6, line 246) and provided justification for the division by 8 (page 7, line 249).
Comments 4 :In the explanation of Figure 2: "According to the initial pheromone distribution rules, pheromone concentration is higher near the line between the starting point and the target point, with a brighter yellow color distribution," the specific positions of the starting point and target point are not clearly marked in the figure. Additionally, the statement "In areas with more obstacles, surrounding pheromones decrease, resulting in lower pheromone concentration and a duller color" is difficult to distinguish clearly due to the indistinct color gradient for regions with varying obstacle densities. The authors should add labels for the start and target points in Figure 2 and enhance the color contrast or provide quantitative annotations to clarify pheromone concentration differences.
Response 4:Thank you for pointing this out. We agree with this comment. Therefore, we have enhanced the colour contrast in Figure 2 and added markers to indicate the starting and ending points. Modifications were also made to the description of the altered image. The modified image is shown in Figure 3 on page 7, with the description of the modified image appearing on lines 255–258 of page 7.
Comments 5:A critical omission is the lack of performance comparison between the fused algorithm and standalone IACO or DWA. The synergistic benefits of the fusion—such as improvements in path length, real-time performance, or obstacle avoidance success rate—are not analyzed.
Response 5:Agreed. Therefore, we have incorporated a performance comparison between the fusion algorithm and the standalone DWA, providing a quantitative analysis. The modifications are detailed in Section 4.3.1, including Figure 18 and Table 8.
Comments 6 :The real-world experimental results are only qualitatively described (e.g., "successful avoidance", "safely reached"). Quantitative metrics—such as path deviation during avoidance, linear velocity fluctuation, response time, path length, and travel time—are missing. Additionally, the paths in Figures 18 and 19 appear to pass close to obstacles, suggesting potential safety risks. The authors should discuss this behavior and supplement the results with quantitative safety measures, such as the minimum distance to obstacles during navigation.
Response 6: Agreed. Therefore, we have supplemented quantitative metrics—such as path deviation during avoidance, linear velocity fluctuations, path length, travel time, and minimum distance from obstacles during navigation. The modifications are detailed in Section 4.3.2, supplemented with both graphical representations and textual descriptions.
Reviewer 2 Report
Comments and Suggestions for Authors
The paper proposes a hybrid global–local path planning method for indoor delivery robots, combining an improved ant colony optimization algorithm with a modified Dynamic Window Approach. The topic is relevant, and the results are promising, but I recommend the following points be addressed before an eventual publication:
- The description of the methodology should be clearer, in particular regarding which measurements are available to the robot and which control variables are used in the global and local planning layers.
- It would help readers to include, early in the paper, a compact overview of the overall control architecture/pipeline, clarifying how the global path planner and local DWA-based planner interact.
- The authors should better highlight the novelty in comparison with existing hybrid ACO–DWA (and related) methods and briefly justify the chosen parameter values (both for IACO and DWA), even if only qualitatively.
- Since the approach targets real-time navigation, the paper should discuss computational cost and expected runtimes on representative hardware.
- The typesetting of the equations must be improved (consistent font sizes, symbols and alignment), and several figures should be enlarged and cleaned up (axes, legends) to improve readability.
Author Response
Comments 1 :The description of the methodology should be clearer, in particular regarding which measurements are available to the robot and which control variables are used in the global and local planning layers.
Response 1:Thank you for pointing this out; we concur with this comment. Consequently, we have incorporated details into the manuscript specifying which measurement data the robot can access, along with the respective control variables employed by the global planning layer and the local planning layer. The revisions are located on page 4 of the manuscript, lines 144–155, and in Figure 1.
Comments 2 :
It would help readers to include, early in the paper, a compact overview of the overall control architecture/pipeline, clarifying how the global path planner and local DWA-based planner interact.
Response 2:
Agreed. Consequently, we have supplemented the overall control architecture with both the global path planner and the local planner. The modifications are depicted in Figure 1 on page 4 of the manuscript.
Comments 3 :
The authors should better highlight the novelty in comparison with existing hybrid ACO–DWA (and related) methods and briefly justify the chosen parameter values (both for IACO and DWA), even if only qualitatively.
Response 3:
Agreed. Therefore, we have incorporated a performance comparison between the fusion algorithm and the standalone DWA, providing a quantitative analysis. The modifications are detailed in Section 4.3.1, including Figure 18 and Table 8.
Comments 4:
Since the approach targets real-time navigation, the paper should discuss computational cost and expected runtimes on representative hardware.
Response 4:
We concur with this comment. As our experiments are simulation-based, we have provided the number of control nodes for path planning and the simulation runtime in Section 4.3.2 (Table 9, page 27). The simulation environment for the experiments is also detailed (lines 492–494, page 15).
Comments 5:
The typesetting of the equations must be improved (consistent font sizes, symbols and alignment), and several figures should be enlarged and cleaned up (axes, legends) to improve readability.
Response 5:
. We agree with this comment. Therefore, we have already revised the overall typesetting of the manuscript, including images and equations.
Reviewer 3 Report
Comments and Suggestions for Authors
In this manuscript the authors present several optimization strategies for hybrid path-planning method for delivery robots that combines an improved Ant Colony Optimization algorithm with an enhanced Dynamic Window Approach to achieve efficient global navigation and robust local obstacle avoidance that significantly improve convergence speed, path smoothness, and stability. The results exhibited through the simulation experiments show that in both static and dynamic scenarios the fusion algorithm outperforms traditional ACO variants and standard planners.
Points that need to be revised:
- The introduction provides sufficient background and relevant references to justify the proposed approach but repeats similar concepts and could be more concise.
- The methods are described in significant detail; however, a few equations and figures need refinement for clarity, but overall, the methodology is well-documented and reproducible.
- The results are clearly presented through tables, figures, and convergence curves, demonstrating improvements over baseline algorithms, however, some figures are difficult to read due to formatting issues, and additional quantitative metrics for dynamic experiments would further improve the analysis.
- Regarding the layout of the manuscript, it is quite important to mention and correct that all figures are repeated twice, one in the text and next as regular figures. Figures and tables convey the necessary information, but several suffer from low resolution, inconsistent numbering, or compressed formatting. The path-planning visuals are informative, yet some axes, labels, and legends could be improved for readability. Tables are generally clear but would benefit from more consistent alignment and spacing. Moreover, some equations are affecting the line spacing.
- Finally, A professional language edit is recommended before publication, as several long sentences could be simplified, and technical descriptions would benefit from more concise expression.
Strengths
- Thoroughly improved ACO framework with innovative heuristics.
- Robust experimental validation concerning sensitivity analysis, the comparison with related approaches and map testing.
- Well-designed fusion strategy that effectively integrates global and local planning.
- The aims are well described regarding the problem motivation as well as their link to real world problems
Weaknesses
- The weaknesses of the manuscript mostly include sentence inconsistencies (proof reading is required), and low resolution figures with inconsistent numbering and insufficient labeling (please remove duplicate figures).
Overall, I think that significant work is presented in the manuscript and the core parts of the research are well discussed and described. I suggest that the manuscript needs a revision concerning the aspects discussed above.
Author Response
Comments 1 :
The introduction provides sufficient background and relevant references to justify the proposed approach but repeats similar concepts and could be more concise.
Response 1:
Thank you for pointing this out. We agree with this comment. Therefore, we have streamlined the similar concepts in the introduction section of the manuscript. (Page 2, lines 57–68).
Comments 2 :
The methods are described in significant detail; however, a few equations and figures need refinement for clarity, but overall, the methodology is well-documented and reproducible.
Response 2:
Thank you for pointing this out. We agree with this comment. Therefore, we have already revised the overall typesetting of the manuscript, including images and equations.
Comments 3 :
The results are clearly presented through tables, figures, and convergence curves, demonstrating improvements over baseline algorithms, however, some figures are difficult to read due to formatting issues, and additional quantitative metrics for dynamic experiments would further improve the analysis.
Response 3:
Agreed. Therefore, we have supplemented quantitative metrics—such as path deviation during avoidance, linear velocity fluctuations, path length, travel time, and minimum distance from obstacles during navigation. The modifications are detailed in Section 4.3.2, supplemented with both graphical representations and textual descriptions.
Comments 4 :
Regarding the layout of the manuscript, it is quite important to mention and correct that all figures are repeated twice, one in the text and next as regular figures. Figures and tables convey the necessary information, but several suffer from low resolution, inconsistent numbering, or compressed formatting. The path-planning visuals are informative, yet some axes, labels, and legends could be improved for readability. Tables are generally clear but would benefit from more consistent alignment and spacing. Moreover, some equations are affecting the line spacing.
Response 4:
We concur with this comment. Consequently, we have revised the overall layout of the manuscript.
Comments 5 :
Finally, A professional language edit is recommended before publication, as several long sentences could be simplified, and technical descriptions would benefit from more concise expression.
Response 5:
Thank you for pointing this out. We have revised certain sentences in the manuscript to enhance conciseness. (Page 4, lines 159–164; Page 7, lines 255–258; Page 26, lines 711–716;)
Round 2
Reviewer 1 Report
Comments and Suggestions for Authors
The paper presents a well-motivated and structured study on hybrid path planning for delivery robots by combining improved ACO and DWA. The revisions, particularly the inclusion of quantitative safety metrics and comparative experiments, have significantly strengthened the manuscript. Overall, the work is valuable for the community and is recommended for acceptance.